# Heterogeneity of Circulating Tumor Cells in Breast Cancer: Identifying Metastatic Seeds

**DOI:** 10.3390/ijms21051696

**Published:** 2020-03-02

**Authors:** Maxim E. Menyailo, Maria S. Tretyakova, Evgeny V. Denisov

**Affiliations:** Laboratory of Cancer Progression Biology, Cancer Research Institute, Tomsk National Research Medical Center, Russian Academy of Sciences, 634009 Tomsk, Russia; max89me@yandex.ru (M.E.M.); trremar@mail.ru (M.S.T.)

**Keywords:** breast cancer, circulating tumor cells, heterogeneity, metastasis

## Abstract

Metastasis being the main cause of breast cancer (BC) mortality represents the complex and multistage process. The entrance of tumor cells into the blood vessels and the appearance of circulating tumor cells (CTCs) seeding and colonizing distant tissues and organs are one of the key stages in the metastatic cascade. Like the primary tumor, CTCs are extremely heterogeneous and presented by clusters and individual cells which consist of phenotypically and genetically distinct subpopulations. However, among this diversity, only a small number of CTCs is able to survive in the bloodstream and to form metastases. The identification of the metastasis-initiating CTCs is believed to be a critical issue in developing therapeutic strategies against metastatic disease. In this review, we summarize the available literature addressing morphological, phenotypic and genetic heterogeneity of CTCs and the molecular makeup of specific subpopulations associated with BC metastasis. Special attention is paid to the need for in vitro and in vivo studies to confirm the tumorigenic and metastatic potential of metastasis-associating CTCs. Finally, we consider treatment approaches that could be effective to eradicate metastatic CTCs and to prevent metastasis.

## 1. Introduction

Breast cancer (BC) is one of the most common types of cancer: it is diagnosed in about 2 million people each year, causing 627,000 deaths annually [1]. The main causes of BC mortality are distant metastases and tumor relapse, as well as the lack of effective treatment and drug resistance [2,3,4].

Metastasis represents a multi-stage process that includes invasion of tumor cells, intravasation into blood vessels with the appearance of circulating tumor cells (CTCs), extravasation into tissues, and formation of micro- and macrometastasis [5]. Cancers can metastasize at an early stage [6]; however, metastasis is more typically observed in advanced cancer [7].

CTCs are heterogeneous and, apparently, not every subpopulation of cells is able to metastasize. Due to the morphological, phenotypic, and genetic plasticity, CTCs adapt to changing environmental conditions [8], modulate therapy efficacy, and demonstrate various ability to metastasize [9]. The study of CTC heterogeneity and identification of metastatic cell types are necessary for the selection of therapeutic strategies to prevent metastatic disease [10,11,12].

Epithelial–mesenchymal plasticity is one of the most important but not the only cause associated with a metastatic phenotype of CTCs [13,14]. As a result of epithelial–mesenchymal transition (EMT), primary tumor cells acquire motility, resistance to apoptosis, senescence, and immune response, as well as drug insensitivity that, in general, increase cell viability in the bloodstream. Nevertheless, the majority of CTCs die, and only some of them extravasate into other tissues, undergo a mesenchymal–epithelial transition (MET), and form metastases [15,16,17]. For example, recent studies described CTCs populations with a hybrid epithelial–mesenchymal phenotype (i.e., with partial EMT) and high metastatic potential [18,19]. In addition to the EMT features, CTCs can also exhibit stem-like characteristics, particularly self-renewal and multilineage differentiation ability [20].

There is still no highly effective therapy for metastatic BC. Therapy targeted directly against CTCs seems to be promising. To date, various molecular markers of CTCs and their therapeutic targets that can be used to prevent cancer progression are known [21]. In addition, different therapeutic approaches for the elimination of CTCs have been proposed; however, none of them has been clinically approved. The reason is that the molecular and biological characteristics of the primary tumor and CTCs are not well understood. In our opinion, the main focus should be on the search for specific populations of CTCs responsible for the development of metastasis and the study of their molecular makeup. These cells could be the primary therapeutic target for preventing metastatic progression [22]. Recent advances in single-cell sequencing allow one to determine the population structure of CTCs and identify metastasis-initiating cells [23,24,25,26], while in vitro and in vivo studies provide data on their proliferative, apoptotic, invasive, and other phenotypes [27].

Nowadays, despite a large number of studies on CTC heterogeneity and their metastatic potential, there are still no generally accepted and universal markers of CTCs prone to metastasis. In this review, we systematized the data on metastatic CTCs in terms of their morphological, phenotypic, and genetic heterogeneity in BC, as well as emphasized the importance of studying these cells in vitro and in vivo and developing therapeutic approaches for their elimination to prevent metastasis.

## 2. CTC Heterogeneity and the Molecular Makeup of Metastatic Cells

CTCs are a small population of cells that enter the bloodstream from the primary tumor and metastases. CTCs are found in most solid tumors, including breast, prostate, lung, bladder, gastric, and other cancers. Several studies reported that CTCs can be observed in non-cancer volunteers [28,29]; however, its number is extremely rare and can be a false-positive rate of the used CTC detection methods.

Successful metastasis depends on the ability of CTCs to adapt, survive and induce neoangiogenesis in the target tissues [8]. Only 2.5% of CTCs form micrometastases and 1% of micrometastases progress to macrometastases [30,31]. The study of CTCs remains a technically challenging task due to the extreme phenotypic heterogeneity and rarity of these cells in the bloodstream and therefore requires the use of highly sensitive and specific methods [32]. Moreover, CTCs are not detected in most of the patients, and when detected, the number of CTCs is usually low—< 5 cells per 7.5 mL [33]. Conclusions on CTC heterogeneity are mostly obtained from patients with the largest numbers of CTCs, which probably have the most aggressive tumors [34].

Identification and isolation of CTCs in BC patients are usually based on the determination of surface epithelial markers, particularly EpCAM and cytokeratins (CK8, CK18, and CK19), and the exclusion of leukocytes by using CD45 [34,35,36]. N-cadherin, vimentin, Twist, Snail, Zeb, and other markers are used for counting CTCs in the EMT state [5]. Such antibody-dependent CTC isolation can be performed using MACS or FACS sorting, as well as CellSearch system and other similar devices. CTCs can be also enriched using polycarbonate track-etched filters [37] and different microfluidic technologies [38,39].

The existence of a large number of methods for CTC analysis indicates how heterogeneous the population of these cells is and how important it is to understand their structure [40,41]. The main problem is the lack of a comprehensive analysis of CTC heterogeneity at the genetic, phenotypic, and morphological levels in terms of identifying the characteristics associated with metastatic progression [42].

### 2.1. Morphological Heterogeneity. Circulating Clusters

Along with individual CTCs, circulating tumor cell clusters (CTC clusters) that represent groups or microemboli (spheroids) consisting of 2–50 cells can be also found in the blood of patients with various malignant neoplasms [43,44].

Similar to CTCs, clusters originating from the primary tumor can return to the original site or enter other organs—potential metastatic sites. Clusters arising from metastases can also return to the primary tumor, the original metastatic site or “self-seed” other tissues and organs [45].

The formation of CTC clusters is assumed to be related to the collective invasion of cohesive groups of tumor cells, passive shedding of tumor clumps at the sites of disrupted endothelium or aggregation of individual tumor cells in migration and circulation [44,46,47]. Clusters circulate in the bloodstream due to their structural deformability [44]. The increased viability of CTC clusters in circulation is probably related to the high production of autocrine pro-migration factors, matrix metalloproteinases, and escape from the immunological surveillance. The metastatic potential of CTC clusters is 23 to 50 times higher as compared to individual CTCs [44]. Nevertheless, in chemotherapy-treated patients with metastatic BC, mortality depends not on the presence of clusters but rather on the number of individual CTCs [48].

CTC clusters appear to originate from oligoclonal groups of tumor cells [44], and their appearance is associated with an increase in the number of mesenchymal CTCs [49]. It has been documented that clusters are more frequent in patients with mesenchymal CTCs than in patients with epithelial CTCs [49]. In breast CTC clusters, tumor cells are interconnected via the adhesion proteins: plakoglobin and CD44 [44,47]. In squamous cell carcinomas, CTC clusters are enriched by another adhesive protein, claudin 11 [50]. It is assumed that increased cell-cell junctions allow the clusters to intravasate and maintain stem-like properties necessary for successful metastatic colonization of distant organs [47,51]. Indeed, knockdown of plakoglobin or CD44 abrogated CTC cluster formation and suppressed metastasis [44,47]. Increased intercellular adhesion and metastatic potential of CTC clusters are also related to the high expression of keratin 14 (K14). Suppression of K14 resulted in the abrogation of distant metastasis, probably through the disruption of the activity of numerous genes, including *TNC* (tenascin C), *JAG1* (Jagged 1), and *EREG* (epiregulin) [46]. In the same study, the authors traced CTC clusters at all of the stages of metastasis: collective invasion, local dissemination, intravasation, circulation, and formation of micrometastases, as well as proved that polyclonal dissemination of CTC clusters is a specific mechanism of BC metastasis (more than 90% of all metastases) [46]. In addition to above-mentioned molecules, CTC clusters were shown to overexpress the transcription factor XBP1, protein disulfide isomerase AGR2, epidermal growth factor receptor HER3, inhibitor of matrix metalloproteinases TIMP-1, plasminogen activator SERPINE1/PAI-1, and antiapoptotic factor BCL2 [44,49,52,53].

In contrast, transcripts encoding classical CTC markers such as keratins, mucin 1 (MUC1), EpCAM, and E-cadherin are underexpressed in CTC clusters. Probably, it indicates a hybrid epithelial–mesenchymal phenotype of the clusters [53]. This EMT state was showed to be associated with poor prognosis in BC patients [18,54].

The DNA methylation landscape of CTC clusters also differs from that in individual CTCs. In particular, clustered cells show hypomethylation of the *OCT4*, *NANOG*, *SOX2*, and *SIN3A* genes involved in the regulation of stemness and proliferation, as well as hypermethylation of polycomb target genes implicated in chromatin remodeling and inhibition of the expression of transcripts responsible for cell differentiation [51].

CTC clusters may contain platelets and immune cells. Such cooperation enhances the viability of tumor cells in the bloodstream [45]. Neutrophils enhance the metastatic potential of tumor cells through overexpression of cell cycle and DNA replication genes. Patients with at least one neutrophil-containing CTC cluster found per 7.5 mL of blood showed significantly worse progression-free survival compared to patients with five or more individual CTCs per the same blood volume [55].

### 2.2. Phenotypic Heterogeneity

CTCs may differ in the ability to proliferate and undergo apoptosis and be heterogeneous in the signature profile of PAM50. CTCs are often triple-negative [56] and negative for Ki-67, which makes them resistant to chemotherapy [57,58]. The apoptotic index (Ki-67^−^/M30^+^) of CTCs increases during clinical dormancy, while the proliferation index (Ki-67^+^/M30^−^) increases on relapse [59]. Expression of the genes involved in cell proliferation (*MYC*, *ATF3*, *TERT*, *RAC1*, *FOXA1*, *RRM1*, *CCNB1*, *BIRC5*, and *Ki-67*) is decreased in CTCs as compared to the primary breast tumor. Probably, this indicates a non-proliferative (dormant) state of CTCs in the bloodstream [60,61]. Cells in this state are believed to have a similar tumorigenic potential, although they demonstrate differences in cellular plasticity, invasion, and metastatic potential [62].

The number of CTCs changes during surgical and therapeutic interventions [20,63,64,65]. Significant changes in the number of stem- and EMT-like CTC populations were observed in BC patients during neoadjuvant chemotherapy [64]. Minor surgical injury (e.g., biopsy) was found to lead to an increase in CTCs without the signs of EMT and stemness (EpCAM^+^CD45^−^CD44^−^CD24^−^N-cadherin^−^) and stem-like CTCs (EpCAM^+^CD45^−^CD44^+^CD24^−^N-cadherin^−^) [20]. Moreover, a significant transcriptional heterogeneity, mainly of the genes encoding EpCAM, HER2/neu, vimentin, and NANOG proteins, is observed in CTCs after surgery. For instance, it has been established that EpCAM is expressed in most of the CTCs, while HER2/neu, vimentin, and NANOG are detected only in some cells [65]. Single-cell sequencing confirmed transcriptional heterogeneity of CTCs in metastatic BC and revealed *MUC16* and *TMPRSS4* genes, the expression of which may be associated with the formation of metastases [66].

EMT is triggered in response to pleiotropic signaling molecules that induce the expression of specific transcription factors (Snail1, Slug, Zeb1/2, Twist 1/2, etc.) and microRNAs (miR200 family) together with epigenetic and post-translational changes. All this ultimately leads to the loss of epithelial markers (E-cadherin, EpCAM, etc.), expression of mesenchymal genes (*CDH2*, *VIM*, etc.), and appearance phenotypic and structural changes associated with increased cell motility and invasiveness [8,67,68]. Increased invasiveness promotes intravasation of tumor cells and ensures their survival in circulation [5,69]. Furthermore, EMT-induced phenotypic changes are associated with acquired stemness, resistance to therapy, and immunosuppression [70]. The reverse process of the EMT, mesenchymal–epithelial transition (MET), leads to a loss of the ability to migrate and restore proliferative and epithelial characteristics necessary for metastatic colonization of distant organs [71]. In recent years, the majority of studies indicate that the phenotype of tumor cells can be “fixed” at the intermediate stages, where the EMT transition is partially accomplished. For this reason, EMT is considered as a continuum in which cells exhibit epithelial, intermediate, and mesenchymal phenotypes [72,73,74,75].

EMT markers are usually found in CTCs of the patients with metastatic BC [76]. Moreover, in ER^+^/PR^+^ BC, CTCs predominantly show the epithelial phenotype, whereas patients with HER2^+^ and triple-negative cancers have mesenchymal CTCs. The number of mesenchymal CTCs increases with the BC progression in response to the treatment [49].

It is clear now that EMT is associated with the increased metastatic potential of CTCs. Upregulation of the genes associated with metastasis (*NPTN*, *S100A4*, and *S100A9*) and EMT (*VIM*, *TGFβ1*, *ZEB2*, *FOXC1*, and *CXCR4*) has been found in CTCs of BC patients and is associated with an unfavorable prognosis [61]. Another study assumed that FOXC1 can contribute to the EMT in CTCs and, as a consequence, to an increase in their metastatic potential [49]. In EpCAM-negative CTCs, HER2^+^EGFR^+^HPSE^+^Notch1^+^ population was identified with the highest potential to metastasize to the brain and lungs [77].

Cancer stem cells present a rare population of tumor cells, which, due to the self-renewal and multilineage differentiation ability, are responsible for tumor initiation and maintenance, and considered a source of metastases [68]. The identification of the stem-like CTCs is based on the assessment of the CD44, ALDH1, and CD133 markers or determination of the associated transcripts [78,79]. As mentioned above, ЕМТ and stemness are closely related to each other. In addition to the acquisition of the migratory and invasive phenotypes, EMT cells express CD44, an antigen characteristic of BC stem cells [80]. High CTC plasticity in terms of EMT and stemness is associated with the poor prognosis and high aggressiveness of BC [81]. The presence of CTCs with a stem-like phenotype correlates with the tumor stage [82] and therapy resistance [76]. CD44^+^CD24^−^ALDH1^+^ CTCs show an increased tumorigenic potential [78], while co-expression of EpCAM, CD44, CD47, and MET is a characteristic of the CTC population with the high metastatic ability [83].

### 2.3. Genetic Heterogeneity

Advances in microarray technology and massively parallel sequencing, including single-cell sequencing, have shown that CTCs, like primary tumor cells, exhibit significant genetic heterogeneity [84,85,86]. However, there is scarce information regarding specific genetic alterations that could be associated with the metastatic potential of CTCs, particularly with an increased ability to migrate, intravasate, change energy metabolism, interact with platelets and immune blood cells, and be resistant to therapy [60,87,88].

Currently available data indicate genetic heterogeneity of the CTCs in the same patient, as well as differences in the mutational landscape between the CTCs and the primary tumor/metastases. These findings indicate that only part of the primary tumor cells has the ability of invasion/intravasation and/or a large percentage of CTCs die in the circulation. 

Targeted sequencing of CTCs and primary tumors showed concordance of *PIK3CA* gene mutations in only 13.73% of the cases [89]. However, the overlapping between CTCs and the primary tumor may depend, among other factors, on tumor heterogeneity. For example, in some cases, the concordance reaches 85% [90].

Similar to the primary tumor, mutational landscape of the CTCs is heterogeneous: Variability has been reported for mutations in the genes *PIK3CA*, *ESR1*, and *KRAS*. Interestingly, some *ESR1* and *KRAS* mutations were present in certain CTCs but were absent in the primary tumor [86]. Numerous DNA copy number aberrations, which are typical for the triple-negative BC, were found in CTCs. These aberrations included amplified chromosome regions characteristic of metastatic BC: 3q, 6p21.2 (*PIM1*), 8q22.1 (*CCNE2*), 8q24.21 (*MYC*), 11q13.3 (*CCND1*), 19p13.2 (*NOTCH3*), 20q13.2 (*AURKA*), as well as 5q12-13 and 16q deletions [91,92,93]. Moreover, amplification of 8q24.21, as well as chromosome 9q in CTCs, was found to be a clonally selected event for the initiation of brain metastasis in BC. It turned out that overexpression of semaphorin-4D (9q) promoted CTCs transmigration through the blood-brain barrier whereas MYC (8q24.21) facilitated the adaptation of tumor cells to the activated brain microenvironment via upregulation of GPX1 enzyme [94].

## 3. In Vitro and in Vivo Study of the CTC Phenotype

To date, CTCs have been identified in many malignant neoplasms. However, their biological characteristics were described mainly using flow cytometry, sequencing, FISH and PCR analysis, spectroscopic technique (SERS), etc. [95,96]. Nevertheless, in vitro and in vivo models can be also used to study the CTC phenotype.

An in vitro study can be useful for the understanding of the proliferative and apoptotic potential of CTCs, their migration and invasion ability, etc. In vivo models can be used to reveal the tumorigenic and metastatic phenotype of various CTC populations. The main difficulty lies in the fact that CTCs in the peripheral blood are represented by a low number of cells, and the technology of CTC transfer to in vitro and in vivo models is laborious [97]. Nonetheless, the development of cell lines from CTCs is still possible. Zhang et al. have established primary cultures from the CTCs isolated from the blood of patients with advanced BC [77]. Subsequent studies generated CTC-derived cell lines from patients with other cancers [98,99,100,101,102]. However, in cell cultures, the availability of oxygen, nutrients, metabolites, and signaling molecules is not limited and thus does not reflect the true picture of tumor growth in a living organism, where interactions not only between the tumor cells but also between the tumor and the surrounding extracellular matrix, as well as stromal and immune cells, play an important role [103].

Tumor spheroids (or tumoroids) are three-dimensional models consisting of tumor cells only or their combination with other types of cells. They simulate intercellular interactions and cell contacts with the environment [104]. Spheroids present a convenient model since they imitate the in vivo characteristics of tumor cells such as growth kinetics, cell heterogeneity, and signaling pathway activity. Spheroids are usually obtained either using scaffolds (matrix-on top, matrix embedded, matrix-encapsulation, spinner flasks, micropatterned plates, ultra-low attachment plates) or without them (hanging drop, magnetic levitation, and magnetic 3D printing) [105]. The first three-dimensional model was obtained by Zhang et al. from CTCs of a patient with lung cancer, and contained fibroblasts and extracellular matrix proteins imitating tumor microenvironment [106].

Organoids are another alternative for obtaining the three-dimensional model of cell cultures. They present structures formed from tissue cells or embryonic/pluripotent stem cells [107]. In general, organoids are obtained in a relatively short time and easily reproduced and stored. They are biologically stable, suitable for screening analyzes, and can be subjected to genetic manipulation [108]. The first organoid lines were obtained from the biopsy and CTCs of prostate cancer patients [109]. 

Despite advantages, 3D models have certain limitations: low uniformity of the spheroids’ size, low efficiency and repeatability, short life span, and work complexity compared to 2D systems. Another problem of using 3D models is the need to introduce components of the tumor microenvironment, mainly stromal and immune cells.

Based on the concept of spheroids and organoids, CTC-derived explant (CDX) models were developed. These models were successfully used for the assessment of tumorigenicity of CTCs isolated from patients with small-cell lung cancer and melanoma [110,111]. However, the role of the immune system is not considered in such models due to the use of immunodeficient mice.

Thus, in vitro and in vivo models of CTCs can be an effective tool in understanding the mechanisms of metastasis in general and revealing the metastatic potential of specific CTC subpopulations, as well as the development of new methods for the prevention of metastatic disease. However, the unpredictable amount of CTCs in the blood of cancer patients is the main factor limiting the widespread use of such models.

## 4. Therapeutic Targeting of Metastatic CTCs

Metastasis remains the leading cause of cancer mortality and is still difficult to treat [112]. The primary cause of unsuccessful anti-metastatic therapy is the significant genetic and phenotypic differences between metastases and the primary tumor. On the one hand, this is due to the independent clonal evolution of metastatic cells, and, on the other, the heterogeneity of CTCs forming secondary tumor foci. In other words, only part of the primary tumor cells enters the bloodstream, with most of the cells dying in the circulation. The remaining CTCs, which reflect the genetic landscape of the primary tumor to a lesser extent, are potentially capable of metastasis formation [15] (Figure 1).

Assessment of metastatic CTCs can be used to predict the probability of metastasis and also for taking necessary diagnostic and therapeutic measures. In addition, the development of novel treatment strategies or drugs based on a molecular portrait of metastatic CTCs seems promising. This would enable the elimination of such tumor cells not so much in the circulation as in the primary site thereby preventing their entry into the bloodstream. To date, there are no antitumor treatment options based on targeting of the metastatic seeds. Nevertheless, various therapeutic methods, which were initially targeted at eliminating the common pool of CTCs but can also be effective against metastasis-initiating cells, have been proposed.

### 4.1. Chemo- and Targeted Therapy Based on Molecular Characteristics of CTCs

It has been proven that anti-HER2 therapy of BC patients with HER2-positive CTCs significantly improves survival [113]. Gefitinib (an anti-EGFR drug) was shown to be highly effective in advanced BC patients with CTCs expressing EGFR [114]. A recent study proposed a multiplex immunofluorescence panel of markers (Top1, Top2, Ki67, RAD51, ABCG2, and γH2AX) combined with TUNEL apoptosis detection, which is focused on the CTCs and designed to predict the response to therapy. The same work has demonstrated that treatment with etirinotecan improves the overall survival of BC patients with low expression of Top1 in CTCs [115]. 

The entrance of tumor cells to the bloodstream and, hypothetically, CTCs themselves can be targeted by “migrastatics” (from Latin ‘*migrare*’ and Greek ‘*statikos*’). This new class of drugs has been proposed to suppress the invasion of cancer cells and, consequently, their ability to metastasize. The most promising agents are multikinase inhibitors targeting either ROCK/MRCK or ROCK/PKA/PKB kinases of the AGC family [116].

Nowadays, dozens of clinical trials evaluating CTCs for prediction of the effect of therapy in patients with different cancers and the effectiveness of various drugs targeting these cells are currently undergoing. For instance, in a multicenter randomized trial (NCT01619111), an analysis of the effectiveness of lapatinib (an anti-HER2/EGFR drug) is performed in metastatic, initially HER2-negative BC patients with HER2-positive CTCs. 

### 4.2. Impairment of CTC Adhesion to Hematopoietic Cells and Platelets

Drugs targeting the adhesion molecules can be used to disrupt the interaction between CTCs and hematopoietic cells. Inhibition of VCAM-1 and ICAM-1 was shown to decrease CTC viability, extravasation, and, as a consequence, their metastatic potential [117]. Disruption of the adhesion between CTCs and platelets using anticoagulants (TFPI, heparin, fucosylated chondroitin sulfate, etc.) enhances the immune-mediated clearance of CTCs, mainly through natural killer cells [118], and results in a more than 80% decrease in metastasis [119,120].

### 4.3. Elimination of CTCs

Advances in the field of nano- and microfluidic technologies triggered the development of devices for CTC elimination and their targeting in vivo. They can be broadly divided into invasive and non-invasive devices. The first ones include various vascular devices that allow continuous screening of CTCs and/or their elimination [121,122]. For instance, a vascular microtube device coated with halloysite and liposomes and containing doxorubicin, ethylene glycol, and E-selectin has been proposed. The use of this device in vitro led to the internalization of liposomes by CTCs and their killing [123]. In another study, a novel biomimetic technique with immobilized E-selectin and apoptosis inducer (TRAIL) has been developed. The use of this device led to the activation of apoptosis in 30% of CTCs after 1 h of treatment [124]. The second class is represented by non-invasive devices (modules) for blood filtering and CTC elimination. An example is immunomagnetic separation devices with antibody-conjugated nanoparticles to CTC markers [125,126,127]. Non-invasive experimental devices also include photoacoustic flow cytometry, which is based on CTC isolation using a complex of nanoparticles and antibodies and their destruction by high temperature generated by photons [128,129,130]. Photodynamic therapy allows selective elimination of CTCs after blood irradiation with a blue laser. The combination of green fluorescent protein (GFP) and photosensitizers induces selective elimination of GFP-expressing CTCs without affecting normal cells [131]. The spaser nanolaser can be used as an optical probe; after its release in the body, it adheres to the CTCs and breaks them into parts [132]. Despite the obvious advantages of these devices, they still have several physiological and anatomical limitations, for instance, susceptibility to infections and thrombosis in the case of invasive devices. In addition, all these devices should be applied continuously because CTCs are supposed to be constantly released from the primary tumor and present in circulation for a short time.

Therapeutic approaches for targeting of CTCs can be much more. For example, Na^+^/K^+^ ATPase inhibitors were recently showed to dissociate CTC clusters into single cells and suppress metastasis [51]. However, none of them have been approved for cancer therapy. Besides the serious limitations mentioned above, the main obstacle that limits the translation of these therapeutic methods is the significant morphological, phenotypic, and genetic heterogeneity of CTCs. One of the potential strategies can be the identification of metastasis-initiating CTCs and their molecular features based on which is possible to optimize the chemotherapy regimen and/or develop new targeted drugs.

## 5. Conclusions

Currently, a number of clinical studies are in progress to use CTCs in different fields of BC management: early diagnosis (including identification of subclinical disease), selection of therapy regimen, evaluation of therapy efficacy, treatment monitoring, and prognosis evaluation. The potential direction in the clinical research of CTCs could be the identification of the metastasis-initiating CTCs and their targeted elimination with the goal of the prevention of metastatic progression. According to current findings, metastatic CTCs in BC have the HER2^+^EGFR^+^HPSE^+^Notch1^+^ phenotype, co-express surface markers EpCAM, CD44, CD47, MET and are characterized by overexpression of the molecules associated with metastasis in general: TMPRSS4, MUC16, S100A4, S100A9, and NPTN. CTC clusters that have more pronounced metastatic potential than individual CTCs exhibit altered activity of the molecules associated with increased cell-cell junctions (plakoglobin and K14), partial EMT (MUC1, EpCAM, and CDH1), and stemness (CD44) and are characterized by high viability due to cooperation with immune cells and platelets. Besides the elimination of CTCs, blocking metastatic progression could be more effective in the case of killing primary cancer cells that are a source of metastasis-initiating CTCs. As suggested by Klotz and colleagues [94], targeting of SEMA4D and GPX1 in primary tumors can be a potential therapeutic approach for the abrogating ability of CTCs to colonize the brain and for preventing brain metastasis in BC patients. This therapy would be an innovative strategy to prevent metastasis at an early stage and even before the tumor is clinically diagnosed.

## Figures and Tables

**Figure 1 ijms-21-01696-f001:**
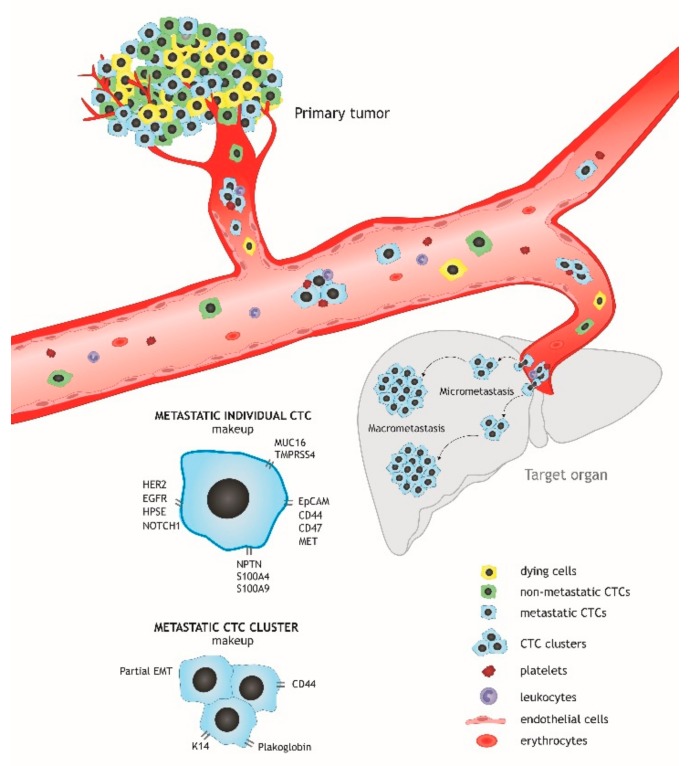
Metastatic circulating tumor cells (CTCs). Most tumor cells entering the circulation die and only part of them survives and is able to extravasate and form micro- and macrometastases. Such cells can be both individual and clustered. Their increased surviving and metastasis-initiating ability is probably related to the expression of specific molecules involved mainly in EMT and stemness maintenance.

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
