# Peer review of "Heterogeneity of Circulating Tumor Cells in Breast Cancer: Identifying Metastatic Seeds"

_ijms, 2020, doi:10.3390/ijms21051696_

Round 1

Reviewer 1 Report

The review of Meniailo and colleagues focuses on circulating tumor cells in breast cancer. The authors introduced the vivid problematics of CTC in diligent way. I can easily imagine that this work can represent valuable introduction into the CTC world for many different newcomers. The authors reviewed the CTC topic in five sections. Following the introduction specifying CTC in breast cancer, they specifically targeted characterization of CTC and their heterogeneity, studies of CTC phenotype, (potential) therapeutic options targeting CTC and concluding remarks. I suppose that the review is fine and I have rather minor comments and suggestions; however, I suppose that they should be considered to increase clarity of the text and better address the current state-of-the-art.

  1. The authors decided to focus on CTC in BC; however many citations are from papers referring about non-BC tumors or in vitro/in vivo models. I do not consider this as a fallacy but the improved description of particular tumor types (especially in sections 3 and 4) enable readers help in better orientation.
  2. I suppose that several concluding statements from cited studies should be modified as they are based on very small samples and not replicated by others. The overall concept of CTC development is still rather hypothesis then proven fact.
  3. The Conclusions should be improved to generalize current state and development of CTC (clinical) research and practical expectations.

Particular notes:

Line 69: However, although rarely, CTCs can be observed in healthy people or people with benign tumors [27]. This statement should be mitigated as follows: Presence of CTCs in non-cancer volunteers is probably extremely rare. In a study by Allard et al CTCs were identified in only one out of 334 analyzed non-cancer subjects.

Line 75: Identification of CTCs is usually based on the determination of… should be as: Identification of CTCs in BC patients is usually based on the determination of….

Line 98: presence of clusters but on the number… should be as: presence of clusters but rather on the number …

Line 100: In patients with only epithelial CTCs, circulating clusters are absent [45]. should be as: It has been documented that clusters are more frequent in patients with mesenchymal CTCs than in patients with epithelial CTC.

Line 136: CTCs are often triple-negative, despite the molecular subtype of the primary tumor, and thus more aggressive [54]. – this speculation should be corrected as the authors of [54] did not correlated phenotypic pattern of CTCs with prognosis in their study. They show that CTCs differ from primary tumors and are more often TNBC.

Line 156- 161. Please, fragment the long sentence.

Line 215: …, 8q4.21 (MYC), should be: …, 8q24.21 (MYC),

Line 258: …immunocompetent mice. should be: …immunodeficient mice.

Line 262: However, the small amount of CTCs… should rather be: However, the unpredictable amount of CTCs…

Line 278-279. Liquid biopsy is not an example of CTC assessment.

Section 4.1. I recommend to add a note about migrastatics as the migrastatic therapy represent an attractive concept that should be mentioned in this review. See e.g. Gandalovicova et al. PMID: 28670628.

Line 297-299: …(an anti-HER2/EGFR drug) is performed in metastatic BC patients with HER2-positive CTCs. Another study evaluates the effect of denosumab on the CTC level in patients with luminal metastatic BC (NCT03070002, 2017-2020). should be added: …(an anti-HER2/EGFR drug) is performed in metastatic, initially HER2-negative BC patients with HER2-positive CTCs. Another study evaluates the effect of denosumab (a RANKL inhibitor) on the CTC level in patients with luminal BC and bone metastases (NCT03070002, 2017-2020).

Line 320: Non-invasive devices also… should be: Non-invasive experimental devices also…

Author Response

Response to Reviewer 1 Comments

Point 1: The authors decided to focus on CTC in BC; however many citations are from papers referring about non-BC tumors or in vitro/in vivo models. I do not consider this as a fallacy but the improved description of particular tumor types (especially in sections 3 and 4) enable readers help in better orientation.

Response 1: First of all, we would like to thank the Reviewer for spending his/her effort and time reviewing our manuscript and providing constructive criticisms. We agree with the Reviewer that the addition of the important points raised will significantly improve our manuscript. The manuscript has been revisited according to the Reviewer’s suggestions. All changes were made using the “Track Changes” function. New references are marked by yellow.

We agree with the Reviewer that there are some citations from non-breast cancer although the manuscript is devoted to CTCs in breast cancer. But most of these are given in the section “In vitro and in vivo study of the CTC phenotype” as a demonstration of the success of obtaining different models from CTCs.

Nevertheless, several references were not related to breast cancer and have been deleted together with corresponding sentences:

Lines 104-106: “Penetration of clusters into the blood vessels is assumed to occur at the sites of disrupted endothelium [40] and/or via tumor cells undergoing EMT [41] and cancer-associated fibroblasts [42], which cause the proteolysis of vessel walls.”

Lines 109-111: “CTC clusters are non-proliferating cells, which makes them resistant to cytotoxic drugs [43]”.

Lines 230-231: “Whole-exome sequencing showed that only 45 of 71 mutations found in CTCs were present in metastases. This once again proves that not all CTCs are able to form metastatic foci [87].”

Additionally, we have replaced two references that were devoted to specific issues in other cancers but were cited in the common context:

Lines 257-259: The main difficulty lies in the fact that CTCs in the peripheral blood are represented by a low number of cells, and the technology of CTC transfer to in vitro and in vivo models is laborious [96].

Lines 270-272: Tumor spheroids (or tumoroids) are three-dimensional models consisting of tumor cells only or their combination with other types of cells. They simulate intercellular interactions and cell contacts with the environment [103].

Point 2: I suppose that several concluding statements from cited studies should be modified as they are based on very small samples and not replicated by others. The overall concept of CTC development is still rather hypothesis then proven fact.

Response 2: We thank the Reviewer for raising this issue. We reanalyzed the key papers focusing on the metastatic phenotype of CTCs and according to them, we made the concluding statements. It turned out that the association of some of them with metastasis was only guessed by the authors. We decided to delete these statements in the conclusion and in the Figure. In the case of CTC clusters, we have added the CD44 as a key factor of their stemness and deleted other proteins: BCL2, OCT4, NANOG, SOX2, and SIN3A. Now, the conclusion about the makeup of individual CTCs and clusters looks as following:

Lines 400-409: According to current findings, metastatic CTCs in BC have the HER2+EGFR+HPSE+Notch1+ phenotype, co-express surface markers EpCAM, CD44, CD47, MET and are characterized by overexpression of the molecules associated with metastasis in general: TMPRSS4, MUC16, S100A4, S100A9, and NPTN. CTC clusters that have more pronounced metastatic potential than individual CTCs exhibit altered activity of the molecules associated with increased cell-cell junctions (plakoglobin and K14), partial EMT (MUC1, EpCAM, and CDH1), and stemness (CD44) and are characterized by high viability due to cooperation with immune cells and platelets.

Point 3: The Conclusions should be improved to generalize current state and development of CTC (clinical) research and practical expectations.

Response 3: We have added the following information: Lines 394-399 “Currently, a number of clinical studies are in progress to use CTCs in different fields of BC management: early diagnosis (including identification of subclinical disease), selection of therapy regimen, evaluation of therapy efficacy, treatment monitoring, and prognosis evaluation. The potential direction in the clinical research of CTCs could be the identification of the metastasis-initiating CTCs and their targeted elimination with the goal of the prevention of metastatic progression.”

Particular notes:

Point 4: Line 69: However, although rarely, CTCs can be observed in healthy people or people with benign tumors [27]. This statement should be mitigated as follows: Presence of CTCs in non-cancer volunteers is probably extremely rare. In a study by Allard et al CTCs were identified in only one out of 334 analyzed non-cancer subjects.

Response 4: We have corrected this part as the Reviewer advised. Also, we have added the second reference (Yap et al., 2019).

Lines 73-75: Several studies reported that CTCs can be observed in non-cancer volunteers [28,29]; however, its number is extremely rare and can be a false-positive rate of the used CTC detection methods.

Point 5: Line 75: Identification of CTCs is usually based on the determination of… should be as: Identification of CTCs in BC patients is usually based on the determination of….

Response 5: We have changed the sentence according to the Reviewer’s suggestion.

Lines 84-86: Identification and isolation of CTCs in BC patients are usually based on the determination of surface epithelial markers, particularly EpCAM and cytokeratins (CK8, CK18, and CK19), and the exclusion of leukocytes by using CD45 [34-36].

Point 6: Line 98: presence of clusters but on the number… should be as: presence of clusters but rather on the number …

Response 6: The sentence has been corrected as the Reviewer suggested.

Lines 114-116: Nevertheless, in chemotherapy-treated patients with metastatic BC, mortality depends not on the presence of clusters but rather on the number of individual CTCs [48].

Point 7: Line 100: In patients with only epithelial CTCs, circulating clusters are absent [45]. should be as: It has been documented that clusters are more frequent in patients with mesenchymal CTCs than in patients with epithelial CTC.

Response 7: The sentence has been corrected.

Lines 118-120: It has been documented that clusters are more frequent in patients with mesenchymal CTCs than in patients with epithelial CTCs [49].

Point 8: Line 136: CTCs are often triple-negative, despite the molecular subtype of the primary tumor [54]. and thus more aggressive. – this speculation should be corrected as the authors of [54] did not correlated phenotypic pattern of CTCs with prognosis in their study. They show that CTCs differ from primary tumors and are more often TNBC.

Response 8: We have deleted the information regarding prognosis that is really not presented in the cited article. The corrected sentence now looks like: Lines 157-159 “CTCs are often triple-negative [55] and negative for Ki-67, which makes them resistant to chemotherapy [56,57].”

Point 9: Line 156- 161. Please, fragment the long sentence.

Response 9: The sentence has been divided.

Lines 178-183: EMT is triggered in response to pleiotropic signaling molecules, that induce the expression of specific transcription factors (Snail1, Slug, Zeb1/2, Twist 1/2, etc.) and microRNAs (miR200 family) together with epigenetic and post-translational changes. All this ultimately leads to the loss of epithelial markers (E-cadherin, EpCAM, etc.), expression of mesenchymal genes (CDH2, VIM, etc.), and appearance phenotypic and structural changes associated with increased cell motility and invasiveness [8,64,65].

Point 10: Line 215: …, 8q4.21 (MYC), should be: …, 8q24.21 (MYC),

Response 10: The misprint has been corrected. Line 244: …, 8q24.21 (MYC),…

Point 11: Line 258: …immunocompetent mice. should be: …immunodeficient mice.

Response 11: Done! Lines 300-301: However, the role of the immune system is not considered in such models due to the use of immunodeficient mice.

Point 12: Line 262: However, the small amount of CTCs… should rather be: However, the unpredictable amount of CTCs…

Response 12: “Small” has been change to “unpredictable”. Lines 305-306: However, the unpredictable amount of CTCs in the blood of cancer patients is the main factor limiting the widespread use of such models.

Point 13: Line 278-279. Liquid biopsy is not an example of CTC assessment.

Response 13: We agree with the Reviewer and have deleted the phrase “(for example, using liquid biopsy)”

Lines 321-322: Assessment of metastatic CTCs can be used to predict the probability of metastasis and also for taking necessary diagnostic and therapeutic measures.

Point 14: Section 4.1. I recommend to add a note about migrastatics as the migrastatic therapy represent an attractive concept that should be mentioned in this review. See e.g. Gandalovicova et al. PMID: 28670628.

Response 14: The information about migrastatics has been added.

Lines 338-342: The entrance of tumor cells to the bloodstream and hypothetically CTCs themselves can be targeted by “migrastatics” (from Latin ‘migrare’ and Greek ‘statikos’). This new class of drugs has been proposed to suppress the invasion of cancer cells and, consequently, their ability to metastasize. The most promising agents are multikinase inhibitors targeting either ROCK/MRCK or ROCK/PKA/PKB kinases of the AGC family [115].

Point 15: Line 297-299: …(an anti-HER2/EGFR drug) is performed in metastatic BC patients with HER2-positive CTCs. Another study evaluates the effect of denosumab on the CTC level in patients with luminal metastatic BC (NCT03070002, 2017-2020). should be added: …(an anti-HER2/EGFR drug) is performed in metastatic, initially HER2-negative BC patients with HER2-positive CTCs. Another study evaluates the effect of denosumab (a RANKL inhibitor) on the CTC level in patients with luminal BC and bone metastases (NCT03070002, 2017-2020).

Response 15: The sentence has been corrected according to the Reviewer’s suggestion.

Lines 345-349: For instance, in a multicenter randomized trial (NCT01619111, 2012-2020), an analysis of the effectiveness of lapatinib (an anti-HER2/EGFR drug) is performed in metastatic, initially HER2-negative BC patients with HER2-positive CTCs. Another study evaluates the effect of denosumab (a RANKL inhibitor) on the CTC level in patients with luminal BC and bone metastases (NCT03070002, 2017-2020).

Point 16: Line 320: Non-invasive devices also… should be: Non-invasive experimental devices also…

Response 16: We corrected the sentence.

Lines 371-373: Non-invasive experimental devices also include photoacoustic flow cytometry, which is based on CTC isolation using a complex of nanoparticles and antibodies and their destruction by high temperature generated by photons [127-129].

Reviewer 2 Report

The topic of this review paper is of a great interest for the field. It attempts to describe the different types of CTCs described and therapeutic strategies. However, the manuscript needs of a major revision. The text is hard to follow, with many large subordinate sentences, and containing many grammatical errors. In addition, most of the statements are an oversimplification of the conclusions reported in the original papers and, in some cases, included in the review as a dogma although they are reported in a single study. Finally, several drawbacks associated with CTC analysis are not discussed.

  1. Line 69: ‘However, although rarely, CTCs can be observed in healthy people or people with 69 benign tumors [27].’ This is an overstatement from the abstract of the paper. According to the manuscript, 1/344 healthy and nonmalignant disase were positive (≥2 CTCs) using CellSearch. This more probably an indicator of the false-positive rate of the technology rather than real CTC or non-diagnosed disease.
  2. Lines 75-82: The authors give the impression that CTC can only be isolated based on the expression of specific markers. However, there are several antigen-independent technologies for CTC isolation based on physical properties if CTCs and depletion of WBC.
  3. Lines 87-89: reseeding and self-seeding from primary tumor or metastasis is not exclusive of CTC clusters.
  4. Lines 92-93: The following statement lacks of a reference ‘Clusters circulate in the bloodstream due to their structural deformability.’
  5. Line 93: As mentioned in lines 124-125, CTC-clusters are not necessarily non-proliferative as shown by their methylation profile.
  6. Line 99: The statement ‘CTC clusters originate from oligoclonal groups of tumor cells [38], and their appearance is associated with an increase in the number of mesenchymal CTCs’ is not accurate. According to ref. 38 CTC clusters are not necessarily oligoclonal and they are not associated with an increased number or M CTCs, but they show an increase of M markers expression.
  7. Line 101: The statement ‘In CTC clusters, tumor cells are interconnected via the adhesion protein plakoglobin. ‘ is not accurate. Other proteins have been proposed (for example Cldn11 and CD44) . Please, review the literature carefully.
  8. Line 128: CTC clusters do not necessarily contain immune cells.
  9. Lines 202-205: The overlapping between the CTCs and primary tumor might depend, among other factors, on the heterogeneity of the primary tumor. In some cases, the overlapping is as larger as 85% (see Paoletti et al., Cancer Research (2018)).
  10. Lines 207-208. This statement have no sense. It gives the impression that CTC were initially supposed to be a homogeneous population across patients and not a reflection of the primary tumor.
  11. CTC are not detected in most of the patients, and when detected, usually the number of CTC is low <5. Conclusions on CTC heterogeneity are mostly obtained from patients with the largest numbers of CTCs, which probably represent the most aggressive tumors. This and other limitations in CTC research must be discussed to a large extend.
  12. Lines 218-219: There are other functional studies performed on CTC using other methods.
  13. Lines 223-225: The main difficulty for establishing cell cultures from CTCs is the low number of cells typically isolated from patients, rather than their ‘individuality’.
  14. Lines 308 – 327: CTCs are supposed to be continuously released from the primary tumor and last for a very limited amount of time in circulation. Under this context, the use of devices that aim to screen and eliminate CTCs should be applied continuously. This limitation is not mentioned in the text.
  15. Line 286: Check also Gkountela et al. (2019) for potential pharmacological approached to target CTC clusters.
  16. It is currently accepted that a large proportion of metastasis occur early in the tumor development (even before the tumor is clinically diagnosed). Under this assumption, the therapeutic targeting of CTCs would have a limited efficacy. Bit of discussion of this issue would be interesting.

Author Response

Response to Reviewer 2 Comments

Point 1: Line 69: ‘However, although rarely, CTCs can be observed in healthy people or people with 69 benign tumors [27].’ This is an overstatement from the abstract of the paper. According to the manuscript, 1/344 healthy and nonmalignant disase were positive (≥2 CTCs) using CellSearch. This more probably an indicator of the false-positive rate of the technology rather than real CTC or non-diagnosed disease.

Response 1: We thank the Reviewer for the time spent in reviewing our manuscript. We appreciate your feedback and have revised the manuscript according to your suggestions. Also, we have edited the English language. All changes were made using the “Track Changes” function. New references are marked by yellow.

Regarding the information about CTCs in healthy people, we have changed the sentence and added the second reference (Yap et al., 2019): Lines 73-75 “Several studies reported that CTCs can be observed in non-cancer volunteers [28,29]; however, its number is extremely rare and can be a false-positive rate of the used CTC detection methods”.

Point 2: Lines 75-82: The authors give the impression that CTC can only be isolated based on the expression of specific markers. However, there are several antigen-independent technologies for CTC isolation based on physical properties if CTCs and depletion of WBC.

Response 2: We agree with the Reviewer. There are different methods for the identification and isolation of CTCs that are based both on antigen-dependent and antigen-independent techniques. We have added some of them in the manuscript: Lines 87-90 “Such antibody-dependent CTC isolation can be performed using MACS or FACS sorting, as well as CellSearch system and other similar devices. CTCs can be also enriched using polycarbonate track-etched filters [37] and different microfluidic technologies [38,39]”.

Point 3: The Lines 87-89: reseeding and self-seeding from primary tumor or metastasis is not exclusive of CTC clusters

Response 3: We agree with the Reviewer. The sentence has been changed and another more relevant reference (Giuliano et al., 2019) has been added: Lines 100-103 “Similar to CTCs, clusters originating from the primary tumor can return to the original site or enter other organs – potential metastatic sites. Clusters arising from metastases can also return to the primary tumor, the original metastatic site or “self-seed” other tissues and organs [45]”

Point 4: Lines 92-93: The following statement lacks of a reference ‘Clusters circulate in the bloodstream due to their structural deformability.’

Response 4: The reference (Aceto et al., 2014) has been added. Line 109: Clusters circulate in the bloodstream due to their structural deformability [44].

Point 5: Line 93: As mentioned in lines 124-125, CTC-clusters are not necessarily non-proliferative as shown by their methylation profile.

Response 5: We have deleted this sentence "CTC clusters are non-proliferating cells, which makes them resistant to cytotoxic drugs” which was focused not on breast cancer. Also, we have changed the reference to Gkountela et al., 2019 in the sentence (lines 144-148) “The DNA methylation landscape of CTC… [50]”.

Point 6: Line 99: The statement ‘CTC clusters originate from oligoclonal groups of tumor cells [38], and their appearance is associated with an increase in the number of mesenchymal CTCs’ is not accurate. According to ref. 38 CTC clusters are not necessarily oligoclonal and they are not associated with an increased number or M CTCs, but they show an increase of M markers expression.

Response 6: We read the paper [38] once again. It really can’t be argued that CTC clusters are oligoclonal. Aceto and colleagues only assume this fact. Regarding the second part of this sentence, we missed the corresponding reference and thus misled you. In reality, Yu et al. showed that the appearance of CTC clusters is related to the number of mesenchymal CTCs. Based on all this, we changed the sentence: Lines 117-118 “CTC clusters appear to originate from oligoclonal groups of tumor cells [44], and their appearance is associated with an increase in the number of mesenchymal CTCs [49]”.

Point 7: Line 101: The statement ‘In CTC clusters, tumor cells are interconnected via the adhesion protein plakoglobin. ‘ is not accurate. Other proteins have been proposed (for example Cldn11 and CD44) . Please, review the literature carefully.

Response 7: We agree with the Reviewer that tumor cells are interconnected via other cell adhesion proteins. In addition to plakoglobin, we have added the CD44 protein and changed references: lines 120-122 “In CTC clusters, tumor cells are interconnected via the adhesion proteins: plakoglobin and CD44 [44,47].” However, we could not find the article which would indicate the involvement of CLDN11 in cell-cell adhesion in CTCs clusters in breast cancer. There is information about the role of this protein in intercellular junctions in CTC clusters of patients with squamous cell carcinoma (Li et al., 2019).

Point 8: Line 128: CTC clusters do not necessarily contain immune cells.

Response 8: The sentence has been changed. Line 149: CTC clusters may contain platelets and immune cells.

Point 9: Lines 202-205: The overlapping between the CTCs and primary tumor might depend, among other factors, on the heterogeneity of the primary tumor. In some cases, the overlapping is as larger as 85% (see Paoletti et al., Cancer Research (2018))

Response 9: We thank the Reviewer for suggesting this important issue. We have added these results: Lines 232-234 “However, the overlapping between CTCs and primary tumor may depend, among other factors, on tumor heterogeneity. For example, in some cases, the concordance reaches 85% [89].”

Point 10: Lines 207-208. This statement have no sense. It gives the impression that CTC were initially supposed to be a homogeneous population across patients and not a reflection of the primary tumor.

Response 10: We agree with the Reviewer that this statement has no important sense. The following sentences have been deleted: Lines 235-238 “Similar to the primary tumor, CTCs mostly contain mutations in the TP53 gene (up to 50% cases) [83,88] as well as in the APC, ERBB2, SMAD4, RB1, CDKN2A, PTEN, and PIK3CA genes [89,90]. In addition, mutations in the ARID1A, CDH1, TTN, RYR2, and LRP2 genes were found in the CTCs; however, their frequency was much lower (1-2 CTCs/1 patient) [90].”

Point 11: CTC are not detected in most of the patients, and when detected, usually the number of CTC is low <5. Conclusions on CTC heterogeneity are mostly obtained from patients with the largest numbers of CTCs, which probably represent the most aggressive tumors. This and other limitations in CTC research must be discussed to a large extend.

Response 11: This is a very valuable addition! We have added this information: Lines 80-83 “Moreover, CTCs are not detected in most of the patients, and when detected, the number of CTCs is usually low < 5 cells per 7.5 ml [33]. Conclusions on CTC heterogeneity are mostly obtained from patients with the largest numbers of CTCs, which probably have the most aggressive tumors.”

Point 12: Lines 218-219: There are other functional studies performed on CTC using other methods.

Response 12: We agree with the Reviewer. There are different methods for CTC analysis. We have added some of them and the corresponding references: Lines 251-253 “However, their biological characteristics were described mainly using flow cytometry, sequencing, FISH and PCR analysis, spectroscopic technique (SERS), etc. [94,95].”

Point 13: Lines 223-225: The main difficulty for establishing cell cultures from CTCs is the low number of cells typically isolated from patients, rather than their ‘individuality’.

Response 13: We thank the Reviewer. The sentence has been corrected: Line 257-259 “The main difficulty lies in the fact that CTCs in the peripheral blood are represented by a low number of cells... [96]”. Also, we have deleted the previous reference and added a more appropriate article by Tellez-Gabriel et al., 2018 [96].

Point 14: Lines 308 – 327: CTCs are supposed to be continuously released from the primary tumor and last for a very limited amount of time in circulation. Under this context, the use of devices that aim to screen and eliminate CTCs should be applied continuously. This limitation is not mentioned in the text.

Response 14: We thank the Reviewer for suggesting this important issue. The corresponding limitation has been added: Lines 378-382 “Despite the obvious advantages of these devices, they still have a number of physiological and anatomical limitations, for instance, susceptibility to infections and thrombosis in the case of invasive devices. In addition, all these devices should be applied continuously because CTCs are supposed to be constantly released from the primary tumor and present in circulation for a short time.”

Point 15: Line 286: Check also Gkountela et al. (2019) for potential pharmacological approached to target CTC clusters.

Response 15: We thank the Reviewer for suggesting this article. The results from this study have added in the manuscript: Lines 383-385 “Therapeutic approaches for targeting of CTCs can be much more. For example, Na+/K+ ATPase inhibitors were recently showed to dissociate CTC clusters into single cells and suppress metastasis [50].”

Point 16: It is currently accepted that a large proportion of metastasis occur early in the tumor development (even before the tumor is clinically diagnosed). Under this assumption, the therapeutic targeting of CTCs would have a limited efficacy. Bit of discussion of this issue would be interesting.

Response 16: We thank the Reviewer for raising this issue. The following information has been added in the Conclusion section: Lines 409-415 “Besides the elimination of CTCs, blocking metastatic progression could be more effective in the case of killing primary cancer cells that are a source of metastasis-initiating CTCs. As suggested by Klotz and colleagues [93], targeting of SEMA4D and GPX1 in primary tumors can be a potential therapeutic approach for the abrogating ability of CTCs to colonize the brain and for preventing brain metastasis in BC patients. This therapy would be an innovative strategy to prevent metastasis at an early stage and even before the tumor is clinically diagnosed.”

Reviewer 3 Report

Authors should add following articles:

1) Marta Tellez-Gabriel, Marie-Françoise Heymann, Dominique Heymann. Circulating Tumor Cells as a Tool for Assessing Tumor Heterogeneity. 2019; 9(16): 4580-4594. doi: 10.7150/thno.34337

2) Jakabova A, Bielcikova Z, Pospisilova E, Matkowski R, Szynglarewicz B, Staszek-Szewczyk U, Zemanova M, Petruzelka L, Eliasova P, Kolostova K, Bobek V. Molecular characterization and heterogeneity of circulating tumor cells in breast cancer. Breast Cancer Res Treat. 2017 Dec;166(3):695-700. doi: 10.1007/s10549-017-4452-9.

3) Manuel Abreu, Pablo Cabezas-Sainz, Thais Pereira-Veiga, Catalina Falo, Alicia Abalo, Idoia Morilla, Teresa Curiel, Juan Cueva, Carmela Rodríguez, Vanesa Varela-Pose, Ramón Lago-Lestón, Patricia Mondelo, Patricia Palacios, Gema Moreno-Bueno, Amparo Cano, Tomás García-Caballero, Miquel Ángel Pujana, Laura Sánchez-Piñón, Clotilde Costa, Rafael López and Laura Muinelo-Romay. Looking for a Better Characterization of Triple- Negative Breast Cancer by Means of Circulating Tumor Cells. J. Clin. Med. 2020, 9(2), 353; https://doi.org/10.3390/jcm9020353

Author Response

Response to Reviewer 3 Comments

Authors should add following articles:

Point 1: Marta Tellez-Gabriel, Marie-Françoise Heymann, Dominique Heymann. Circulating Tumor Cells as a Tool for Assessing Tumor Heterogeneity. 2019; 9(16): 4580-4594. doi: 10.7150/thno.34337

Response 1: We thank the Reviewer for suggesting interesting articles. We added this one to the sentence “The study of CTC heterogeneity and identification of metastatic cell types are necessary for the selection of therapeutic strategies to prevent metastatic disease [10-12]” (lines 38-39).

Point 2: Jakabova A, Bielcikova Z, Pospisilova E, Matkowski R, Szynglarewicz B, Staszek-Szewczyk U, Zemanova M, Petruzelka L, Eliasova P, Kolostova K, Bobek V. Molecular characterization and heterogeneity of circulating tumor cells in breast cancer. Breast Cancer Res Treat. 2017 Dec;166(3):695-700. doi: 10.1007/s10549-017-4452-9.

Response 2: This article has been cited in the sentence “The number of CTCs changes during surgical and therapeutic interventions [20,62-64]” (lines 166-167).

Point 3: Manuel Abreu, Pablo Cabezas-Sainz, Thais Pereira-Veiga, Catalina Falo, Alicia Abalo, Idoia Morilla, Teresa Curiel, Juan Cueva, Carmela Rodríguez, Vanesa Varela-Pose, Ramón Lago-Lestón, Patricia Mondelo, Patricia Palacios, Gema Moreno-Bueno, Amparo Cano, Tomás García-Caballero, Miquel Ángel Pujana, Laura Sánchez-Piñón, Clotilde Costa, Rafael López and Laura Muinelo-Romay. Looking for a Better Characterization of Triple- Negative Breast Cancer by Means of Circulating Tumor Cells. J. Clin. Med. 2020, 9(2), 353; https://doi.org/10.3390/jcm9020353

Response 3: This article has been cited in the sentence “High CTC plasticity in terms of EMT and stemness is associated with the poor prognosis and high aggressiveness of BC [80]” (lines 213-214).

Round 2

Reviewer 2 Report

The authors have addressed my questions appropriately. I only have two minor comments before publication

  • Previous point 7: the reference was Li, C.F., et al., Snail-induced claudin-11 prompts collective migration for tumour progression. Nat Cell Biol, 2019. 21(2): p. 251-262
  • Previous point 11: Lines 76-78. The sentence 'Conclusions on CTC heterogeneity are mostly obtained from patients with the largest numbers of CTCs, which probably have the most aggressive tumors.' needs a reference. I suggest PMID 15317891

Author Response

Response to Reviewer 2 Comments (Round 2)

Point 1: Previous point 7: the reference was Li, C.F., et al., Snail-induced claudin-11 prompts collective migration for tumour progression. Nat Cell Biol, 2019. 21(2): p. 251-262

Response 1: We thank the Reviewer. Because claudin 11 was found to be expressed in CTC clusters in squamous cell carcinoma, but not in breast cancer, we modified the text in the following manner: Lines 109-111 “In breast CTC clusters, tumor cells are interconnected via the adhesion proteins: plakoglobin and CD44 [44,47]. In squamous cell carcinomas, CTC clusters are enriched by another adhesive protein, claudin 11 [50].”

Point 2: Previous point 11: Lines 76-78. The sentence 'Conclusions on CTC heterogeneity are mostly obtained from patients with the largest numbers of CTCs, which probably have the most aggressive tumors.' needs a reference. I suggest PMID 15317891

Response 2: We thank the Reviewer. The reference has been added: Lines 76-77 “Conclusions on CTC heterogeneity are mostly obtained from patients with the largest numbers of CTCs, which probably have the most aggressive tumors [34].”

Reviewer 3 Report

The authors included my suggestions, to me this improve the quality of the article.

Author Response

We thank the Reviewer!